# Habituation of the Light-Startle Response of Orange Head Cockroaches (*Eublaberus posticus*): Effects of Acclimation, Stimulus Duration, Presence of Food, and Intertrial Interval

**DOI:** 10.3390/insects12040339

**Published:** 2021-04-11

**Authors:** Christopher A. Varnon, Ann Taylor Adams

**Affiliations:** 1Department of Psychology, Converse College, Spartanburg, SC 29302, USA; 2Department of Child and Family Studies, University of South Florida, Tampa, FL 33620, USA; annadams@mail.usf.edu

**Keywords:** cockroach, Blattodea, Blaberidae, *Eublaberus posticus*, learning, conditioning, habituation, comparative psychology, behavioral ecology, behavior analysis

## Abstract

**Simple Summary:**

This paper investigates habituation of the light-startle response (LSR) in orange head cockroaches (*Eublaberus posticus*) to improve our understanding of comparative insect psychology. Across four experiments, we found that cockroaches quickly learned to respond less to sudden changes in lighting. We also documented a number of findings common to habituation research, connecting our results to those of other popular model organisms. Our work lays a strong foundation for future research on the behavior of orange head cockroaches as well as learning in cockroaches in general.

**Abstract:**

The purpose of this paper is to establish the orange head cockroach (*Eublaberus posticus*) as a useful insect subject for research in comparative psychology by investigating habituation of the light-startle response (LSR). While one goal of comparative psychology is to compare the behavior of a diversity of species, many taxa, including cockroaches, are grossly underrepresented. Our work serves to improve this deficit by investigating habituation learning in the orange head cockroach in four experiments. In our first experiment, we found that LSR, and habituation of LSR, occurs to both lights being turned on and lights being turned off. In our second experiment, we found that the duration of a light did not affect response, and that spontaneous recovery of LSR occurs after 24 h intervals. In our third experiment, we found that the presence of food inhibited LSR. In our final experiment, we found that the rate of LSR habituation decreased as intertrial interval increased, in a manner predicted by established principles of habituation. Our work lays a strong foundation for future research on the behavior of orange head cockroaches as well as learning in cockroaches in general. We hope that our findings help establish cockroaches as practical insect subjects for research in comparative psychology and related fields such as behavior analysis and behavioral ecology.

## 1. Introduction

The purpose of this paper is to establish the orange head cockroach (*Eublaberus posticus*) as a useful insect subject for research and teaching in comparative psychology by investigating habituation of the light-startle response. The field of comparative psychology, now sometimes called comparative cognition, is an interdisciplinary field between biology and psychology influenced by a number of historical figures, including Aristotle, Charles Darwin, Nikolaas Tinbergen, and B. F. Skinner [1,2,3,4,5]. Comparative psychology primarily seeks to explore similarities and differences in psychological processes across species, then, as a secondary goal connected to other fields, aims to understand why such similarities and differences exist following the general framework of Tinbergen’s four causes of behavior [6]. Unfortunately, the field of comparative psychology has a century-long criticism of over-relying on a few popular species and ignoring a vast diversity of taxa; initially, focusing on rats [2,7,8], and later, non-human primates [9,10]. Most recently, Varnon et al. [10] also reported species that appeared in at least 10 of their 1912 reviewed experiments and found that while 18 mammals and 8 birds occurred in at least 10 experiments, only a single insect (honey bees) was represented in at least 10 experiments. The lack of species diversity in psychology research with respect to insects is concerning. Additional research could be especially useful given that complex insect behavior occurs with a different nervous system organization from the majority of taxa studied. Indeed, there may be some fundamental differences in behavior and learning between vertebrates and insects [11]. We believe that cockroaches are an ideal taxa to fill some of these gaps in the comparative psychology literature, and may also provide an excellent complement to the bee dominated insect psychology literature due to their vastly different behavioral ecology. For example, while honey bees are a eusocial species with a highly specialized diet, cockroaches are social generalists. In addition to providing contrast with bee psychology literature, the social generalist nature of cockroaches may also be a better point of comparison to other social generalist species, including humans and other popular models such as rats.

This paper specifically seeks to study habituation of the light-startle response of *E. posticus*. We selected this species as a laboratory model because it cannot fly or climb smooth surfaces, it is a Central and South American species that would have difficulty infesting a facility outside of a tropical environment, and despite being easy to handle it is fairly active with a voracious appetite, even predating on smaller insects [12]. While there are some studies of their social behavior [13,14], we have not seen research on any startle responses or any form of learning in this species.

We begin our investigations of learning in *E. posticus* by studying habituation, one of the simplest forms of learning. Habituation is defined as a reduction in response to a stimulus as a function of repeated presentations of that stimulus [15,16]. This form of conditioning can be observed across a wide range of animals, from bees [17], to rattlesnakes [18], to rodents [19]. While there is not an abundance of literature on habituation of startle responses in cockroaches, there is sufficient literature to suggest the possibility. The startle response of *Periplaneta americana* is well studied, and occurs in response to multiple types of stimuli [20,21,22,23,24]. Similar startle responses in *Blaberus craniifer* have been shown habituate [25,26]. Additionally, the disturbance hiss of *Gromphadorhina portentosa* can habituate to human handlers [27].

Our behavior of interest is the light-startle response (LSR). When encountering a rapid change in lighting, such as when the laboratory lights are turned on or turned off, *E. posticus* often responds by running forward erratically and then may burrow under substrate. While most cockroach startle research focuses on delivering a puff of air to the cerci, a number of experiments show that innate responses to light may be altered through learning. For example, while *P. americana* naturally avoids light [28], they can also be trained to lift a leg to deactivate light [29], and can even be trained to avoid dark areas [30]. *P. americana* also appears able to use light as a cue in more complex associative learning procedures [31]. Although startle responses as well as innate and learned responses to light have been studied in some species, there is little information about habituation of visually induced responses.

In this paper, we discuss habituation of LSR in *E. posticus* across four experiments. In our first experiment, we investigate LSR in the context it is frequently observed in a laboratory setting as the room lights are turned on and off. In our second experiment, we explore whether LSR is affected by the duration of a flash of light, and whether any habituation learning is retained across multiple days of the experiment. In our third experiment, we explore whether the initial level of LSR or habituation of LSR is affected by the presence of food. In our fourth experiment, we explore whether the rate of LSR habituation is affected by the time between stimulus presentations, as is a common finding in habituation research. Finally, we discuss implications for future research with the *E. posticus*.

## 2. Experiment 1

In our first experiment, we investigated LSR in the context we initially encountered it. In our observations, *E. posticus* is somewhat active both during the day and at night, with adults and large nymphs commonly found on the surface or climbing, and smaller nymphs primarily burrowed in the substrate. This spatial division of adults and nymphs is not uncommon in related (*Blaberus*) cave-dwelling species [32]. A toggle of the room lights often sent a number of individuals audibly scurrying about. This response occurred both when the lights were turned on, and when the lights were turned off. In this experiment, we explored this response by bringing it into a controlled experimental setting. Additionally, we also investigated the effect of acclimation to the experimental apparatus before beginning habituation trials, in order to determine optimal procedures for future research.

### 2.1. Methods

#### 2.1.1. Subjects

Orange head cockroaches (*n* = 96) were used as subjects for this experiment. We selected adult cockroaches that had intact antennae and legs, and at least half of their wings, and had not recently molted. Prior to the experiment, cockroaches lived in large breeding colonies (52 × 36 × 36 cm), on a layer of Repti Bark substrate (Zoo Med Laboratories; San Luis Obispo, CA, USA). Founding members of the colony were obtained from Roach Crossing (roachcrossing.com, accessed on 11 July 2016), Josh’s Frogs (joshsfrogs.com, accessed on 2 March 2020), and Cape Cod Roaches (capecodroaches.com, accessed on 03 July 2020). Cockroaches were provided with dry dog food, produce and water ad libitum. The colony was placed on a shelf unit above other heated enclosures, causing the colony to maintain a temperature of approximately 23 °C. The colony was maintained on a 12:12 h day:night cycle. After participating in the experiment, subjects were placed in a second colony reserved for experimentally experienced roaches. Cockroaches in the colonies exhibited a wide range of natural feeding, territorial and reproductive behaviors.

#### 2.1.2. Procedure

Adult cockroaches were collected in sets of 12, weighed, and then placed individually in small plastic bins (14 × 12 × 7.5 cm). The 12 bins were then placed in individual cells in a large cardboard apparatus. Each subjects’ cell was shielded from any light being presented in other cells by cardboard walls, with thick layers of black duct tape used to cover any gaps. Subjects were allowed to acclimate to the apparatus for either 1 or 24 h, depending on experimental assignment. During this period, the four fluorescent room lights (32 watts, 2196 lumens each) were either turned off, or left on, also depending on experimental assignment. To minimize the effect of other stimuli, the laboratory was completely unused during this period. 

After the acclimation period ended, each subject experienced eight 10-s conditioning trials separated by 30-min intertrial intervals (ITI). During trials, the room lights were toggled on or off; one toggle for each trial. For subjects that received an acclimation period with the room lights on, the first trial involved turning the room lights off, while the second trial involved turning the room lights on. Conversely, for subjects that received an acclimation period with the room lights off, the first trial involved turning the room lights on, while the second trial involved turning the room lights off. These trial types will be hereby referred to as light-on and light-off trials. The alternating patterns were repeated for the eight conditioning trials, resulting in four light-on trials and four light-off trials for each subject. During all trials, two 3-watt red light bulbs (Feit Electric; Pico Rivera, CA, USA) were used to provide illumination for the experimenters. Red light was used as it does not disrupt the behavior of insects as they cannot see red light. The subjects’ behavior in response to room light-changes was filmed during the 10-s trials and scored after the experiment was complete. A camera was mounted approximately 1.2 m above the apparatus, and was set to record for one minute before and one minute after each trial. During trials, an experimenter toggled the room light-switch, which was 2.5 m away from the apparatus. For each trial, we recorded the presence or absence of a startle response. We defined a startle response as any change in location inside the apparatus during the trial. If the subject was already moving when the trial began, a startle response could not be recorded. While we did notice changes in antenna activity, primarily a cessation of antennae movement on stimulus presentation, we could not reliably record this response and thus only defined the startle response in terms of change in location of subjects. After the final trial, we also recorded the presence of startle response as the subjects were recollected to determine whether subjects showing a startle decrease during the experiment were habituating to the light stimuli or were instead displaying fatigue. For the duration of the conditioning trials, including both the trials and the ITI, no activity was permitted in the laboratory aside from the careful movement of the experimenters.

#### 2.1.3. Analysis

All analyses were conducted through the StatsModels package [33] included in the Anaconda distribution of Python, a free scientific analysis distribution of the Python programming language. We analyzed probability of startle using a repeated measures regression via generalized estimating equations (GEE) with a logistic link [34]. The repeated measures aspect of the GEE regression controlled for repeated measures from each subject across the eight conditioning trials using an exchangeable dependence structure. Note that for the logistic regressions, the parameter estimates represent the log odds of a response occurring. While the final logistic regression prediction can be transformed into a probability value, the direction and magnitude of the parameter estimates can also be easily interpreted without transformation. Positive estimates indicate an increase above 50% while negative estimates indicate a decrease below 50%. Absolute value represents the magnitude of the effect. Parameter estimates were compared directly using a z score created by dividing the difference between the estimates by the square root of the sum of the squared standard errors of the estimates [35,36]. See [37] for a detailed discussion on GEE with a focus on using logistic links to study behavior.

### 2.2. Results

Figure 1 shows the probability of startle during the experiment, divided by acclimation lighting (rows) and by acclimation time (columns). While each subject experienced eight trials (four of each type), Figure 1 shows the types of trials as separate lines. When considering the probability of startle when the acclimation light was off (bottom row), clear trends are evident. Consistent decreases in startle probability across trials were observed. As all subjects showed startle responses during recollection and were able to walk immediately after the experiment was complete, this decrease in startle response was likely due to habituation, not fatigue. The habituation trends can be seen both in light-off and light-on trials, though the overall probability of startle was much higher in light-off trials. It appears that the probability of startle response was unaffected by acclimation period. When considering the probability of startle when the acclimation light was on (top row), the trends were less clear. While some overall decrease across trials can be observed, this effect is not pronounced. Additionally, no clear differences were observed between light-off and light-on trials. Again, no differences were observed when comparing the 1-h and 24-h acclimation periods.

Table 1 shows a regression analysis of the results. We included all experimental parameters except acclimation time as the graphs and exploratory analysis indicated little effect (*p* > 0.144). We also included weight as a factor. The intercept for this analysis includes the probability of startle when trial is 0, the trial type is light-off, the acclimation light is off, and weight is 0 g. The analysis shows that the overall response probability was lower for light-on trials (estimate = −1.918, *p* = 0.005), and that response probability was lower for subjects that had a light-on acclimation period (estimate = −2.839, *p* < 0.000). However, there was not a significant combined effect for the light-on trials for subjects that received a light-on acclimation. The trial factor for this analysis shows how the probability of response changes across trials, assuming the trial type is light-off, the acclimation light is off, and weight is 0 g. The decrease in response probability across trials was significant (estimate = −3.09, *p* < 0.000), and the trial interactions reveal that this decrease was similar across all combinations of experimental parameters (*p* values > 0.251). Taken together, the analysis supports what can be seen in the graph and shows that while the overall probability of response may be initially higher or lower for some conditions, the response habituates at a similar rate. Interestingly, the analysis also shows that startle probability somewhat increased with weight (estimate = 0.411, *p* < 0.003).

### 2.3. Discussion

This was the first experiment to show habituation, or any form of learning, in *E. posticus*. The fact that LSR decrease occurred across alternating light-off and light-on trials, as well as the high level of response when recollected makes us confident that we were observing true habituation learning, not sensory adaptation or fatigue. It is interesting that the acclimation lighting had such a pronounced effect on the overall startle behavior. One possible explanation is that individuals that received a dark acclimation period may have been in the active, dark phase of their circadian rhythm. Both *Blatta orientalis* and *P. americana* show peaks in activity immediately after the dark phase begins with *B. orientalis* also showing reduced responses to stimuli in the light phase [38,39]. In some cases, these circadian rhythm effects may extend not only to activity, but also to learning and memory. Eiserer and Ramsay [40] found improved performance in a light-escape task after prior exposure to a dark period in *P. americana*. Additionally, in a series of studies, Page et al. found that in *Rhyparobia maderae* acquisition and recall may inhibited during light portions of the circadian rhythm, depending on the specific learning task, and these effects may be regulated by the optic lobe [41,42,43]. In our case, however, the statistical analysis suggests that only the overall probability of startle was affected, not the rate of habituation. Instead, the rate at which our *E. posticus* learned was similar across all combinations of experimental conditions. The fact that the duration of the acclimation light had no effect on response patterns is also in line with findings suggesting that *B. orientalis* rapidly adapts to changes in circadian cycles [38]. 

Regardless of acclimation lighting and duration, our subjects responded more in light-off trials. While there is little literature to suggest a reason for this difference in startle behavior, it is reasonable to imagine how any immediate change in lighting may signal potential predation. For example, an immediate switch to darkness could occur as a predator approaches and occludes a cockroach, while an immediate switch to light could occur as a predator uncovers a cockroach by moving an object the cockroach was resting under. Generally, our anecdotal observations match the findings of this experiment; in our laboratory colonies, *E. posticus* appears to startle more in response to turning off lights than turning on lights.

## 3. Experiment 2

Our second experiment had several goals. First, although our initial experiment showed that more responses occurred in light-off trials, we wanted a more practical way of precisely delivering a light-change stimulus, and thus tested responses to a burst of light delivered from a hand-held flashlight. Second, we wanted to determine whether the duration of light had any effect on behavior. Generally, stronger stimuli produce more responses and a slower rate of habituation [15,16]. We considered the possibility that longer durations of light might act as more intense stimuli and therefore produce more startle responses and slower habituation rates. Finally, we wanted to determine whether LSR habituation was retained on a second day of habituation training, or whether any relearning effects occurred. Generally, when habituated responses are not fully retained, subsequent rehabituation may occur at a higher rate than the initial habitation [15,16]. While there are several studies on long-term memory in cockroaches [44,45,46], we are not aware of any studies on retention or reacquisition of habituation in cockroaches.

### 3.1. Methods

The general maintenance and collection of subjects, as well as the apparatus were the same as described in experiment 1. Subjects were collected in sets of 12 and were allowed to acclimate in the experimental apparatus for one hour, with the room lights off and the red lights on. After the acclimation period was complete, each subject experienced five conditioning trials with a 30-min ITI. During conditioning trials, each subject was individually presented with a flash of light, delivered by a 100-lumen flashlight held approximately 15 cm above the subject, for 2, 4, 8 or 10 s. The flashlight was placed into each subjects’ cell in the cardboard apparatus so that the light affected only the intended subject. After the five conditioning trials were complete, subjects remained in the apparatus, with the room lights off, for an additional 24 h, and then participated in five additional conditioning trials. Each subject, therefore, experienced 10 trials total across 2 consecutive days. A total of 60 subjects were used, 15 for each light duration. During each trial, we observed subjects for 10 s, starting with the onset of the light. The 100-lumen flash of light appeared much brighter in initial camera recordings than the room lights from the first experiment, prohibiting filming of these conditioning trials. Instead, experimenters immediately recorded the presence of startle after each subject’s trial. As with the previous experiment, no activity in the laboratory was permitted during the acclimation period, and only the minimal activity of the experimenters was present during conditioning trials. Data was analyzed using a similar GEE regression analysis as in experiment 1.

### 3.2. Results

Figure 2 shows the probability of startle for both days of the experiment. Though the data is variable, an overall decrease in probability of startle can be observed across trials for both days. As all subjects startled during recollection and were observed moving immediately after the experiment was complete, it is likely that this decrease was caused by habituation, not fatigue. There did not appear to be any clear differences between response patterns across days that would suggest retention. Light duration did not appear to have any impact on startle probability, suggesting the startle response is caused by discrete changes in lighting, not the continuous presence of an aversive light. 

Table 2 shows a regression analysis of startle probability. We included all experimental parameters in the model, and also included weight as a factor. Generally, the findings support what can be observed in the graphs. Startle probability decreased across trial (estimate = −0.222, *p* = 0.004). Day did not have a significant effect (*p* = 0.400), suggesting that the initial level of response was similar between both days. The interaction between trial and day was also not significant (*p* = 0.083), and while the *p* value is much smaller here, this suggests that a reacquisition effect is unlikely. Light duration also did not have a significant effect, and in this case, there was no effect of weight.

### 3.3. Discussion

The overall rate of habituation in this experiment appeared similar to that of the previous experiment. The primary difference was that, in this experiment, we were able to establish a high level of response to a light-on stimulus. This is likely possible because the light-on stimulus was presented close to the subjects. Although we again observed habituation of LSR, light duration did not appear to have an effect. Instead, it appears that LSR is affected only by the onset of a light change. 

Interestingly, we observed no clear retention or relearning effects. This is in contrast to the established principles of habituation [15,16]. However, a number of other experiments report similar absences of retention. Davis [47] found an inverse relationship between initial rate of startle habituation and retest performance in rats; subjects that experienced lower ITIs showed faster habituation, but worse retention. Similar findings have since been reported for swimming responses in nematodes [48], gill withdrawal responses in sea hares [49], visual startle responses of crabs [50,51], exploratory responses of rodents [52,53], and orientation responses of humans [54]. Wagner [55] suggests that short-term and long-term habituation may be separate memory processes, with long-term habituation actually being an associative learning process. It is interesting to note that research reporting long-term memory in cockroaches have used traditional associative learning procedures [44]. It is possible that the absence of retention or relearning effects in our experiment may have occurred because the ITI was too low to facilitate development of long-term memory using an associative habituation process.

## 4. Experiment 3

In our third experiment, we wanted to determine whether the presence of food had any effect on LSR. It is reasonable to question whether cockroaches kept in an apparatus for a long period of time, without access to food or water, would show decreased responses simply due to fatigue. While our second experiment suggested no fatigue effect between days, investigating the impact of continuous access to food could be beneficial for long term experiments. We therefore wanted to test whether it was possible to provide nutrition inside the experimental apparatus without disrupting behavior. The literature reports mixed findings on this topic. Although some rodent research suggests that habituation of startle responses is not affected by satiation or deprivation [56], startle responses of rats can also be reduced by the presence of stimuli previously associated with food [57], and, in humans, startle responses can be inhibited by engaging in other activities [58,59]. With respect to the general effects of deprivation in cockroaches, the results are also mixed. Reynierse, Manning & Cafferty [60] found that *Nauphoeta cinerea* becomes inactive when deprived of food and water; however, Barcay and Bennet [61] found that adult *B. germanica* males, but not females, became more active after deprivation. For our work, it is possible that the presence of food could be either beneficial or detrimental to the procedure.

### 4.1. Methods

In this experiment, after collection and a 1-h dark acclimation period, subjects received three trials consisting of 10-s, 100-lumen light presentations, separated by 90-min intertrial intervals. The general methods followed those of experiment 2. Subjects were assigned to either a control group (*n* = 22) or a food group (*n* = 22). Subjects in the control group were treated similarly to those in previous experiments, while subjects in the food group also received a small piece of apple inside their individual plastic bins. For this experiment, all subjects were weighed both before and after the conditioning procedure. Experiments recorded the presence of startle during each trial. Data was analyzed using a similar GEE regression analysis as in experiment 1. 

### 4.2. Results

Figure 3 shows the probability of startle for both groups. The habituation curve observed for the control subjects was similar to the curves observed in previous experiments. As before, all subjects responded to recollection procedures and were able to move immediately after the experiment. Interestingly, for the food group, a steady increase in response was observed. Table 3 shows a regression analysis of startle probability. We included all parameters and pre-experiment weight as factors in the model. Here, we used an interceptless model where the parameters for each group are treated as mutually exclusive. Comparing the initial values of the control to the food group reveals no significant difference (estimate difference = 2.050, z = 1.193, *p* = 0.233). The control group showed a significant decrease in response across trial (estimate = −0.704, *p* = 0.010), and while the food group did not show a significant increase across trial, the difference between the two slopes was significant (estimate difference = −1.092, z = −2.874, *p* = 0.004). Weight again appeared to have no effect on probability of startle. 

We compared weight change before and after the experiment to determine whether the subjects in the food group were actually consuming the apple, as it appeared that differences in startle probability were caused by the food group’s ability to eat during the procedure. Subjects in the control group lost an average of −0.035 g (−1.088% change) across the acclimation period and three trials of the experiment (four hours), likely due to waste. Subjects in the food group, however, gained an average of 0.014 g (0.275% change) during the experiment. An independent samples *t*-test revealed the difference in weight change between groups was significant (*t*(42) = 2.463, *p* = 0.018).

### 4.3. Discussion

Subjects in the control group showed similar habituation trends to the previous experiments. Subjects in the food group, however, showed the opposite trend, suggesting that, for our work, providing food during experiments is detrimental. It may require a much longer time frame before providing food and water during an experiment is beneficial. Reynierse et al. [60], for example, did not observe any deaths in *P*. *americana* until 46 days of deprivation. In learning experiments spanning multiple weeks, it may indeed be beneficial to provide food and water. However, access to nutrients should be provided between trials, not during trials. Our findings also suggest that LSR can not only decrease as a function of habituation learning, but can also be inhibited in certain situations. In addition to food, inhibiting stimuli may include odors related to predation, reproduction, or aggregation [48].

## 5. Experiment 4

In our final experiment, we explored the effect of ITI on the rate of habituation. Generally, the longer the ITI, the slower the rate of habituation [15,16]. In addition to determining whether this established effect occurs in cockroaches, we also hoped to identify the ITI where habituation no longer occurs.

### 5.1. Methods

After collection and a 1 h dark acclimation period, subjects received three trials consisting of 10-s, 100-lumen light presentations. The general methods followed those of experiments 2 and 3. Subjects received an intertrial interval (ITI) of either 5, 20, 90, 360, or 1140 min as a function of group assignment, with 20 subjects per ITI group. Subjects were run in sets of 10, with all 10 of each set belonging to the same ITI group. Data was recorded and analyzed in a similar manner to previous experiments.

### 5.2. Results

Figure 4 shows the probability of startle in trials 1, 2 and 3 as a function of ITI. While startle probability remains fairly consistent for the first trial, it increases with ITI for trials 2 and 3. Aside from the 1440-min ITI, startle probability also decreases across trials, suggesting habituation was occurring. It does not appear that any habituation occurs with an intertrial interval of 1440 min. Table 4 shows accompanying statistical analysis that supports the findings of the graph. While startle probability significantly decreases across trial (estimate = −0.838, *p* = 0.000), this effect is significantly dampened by ITI (estimate = 0.001, *p* = 0.004). Although the Trial × ITI estimate is small (0.000673), it is important to remember that this estimate represents the effect of increasing ITI by a single minute. Weight did not appear to have a significant impact on probability of startle, though the *p* value (0.054) was much smaller than in experiments 2 and 3.

### 5.3. Discussion

Our final experiment again shows a familiar pattern of habituation across trials, and also is the first to confirm that shorter ITIs facilitate habituation learning in cockroaches as suggested by the principles of habituation outlined in Thompson and Spencer [15,16]. While this is an established finding in many vertebrate taxa [47], invertebrate research has more often focused on the effect of trial spacing in associative learning, such as how trial spacing affects recall of odor-food associations [62,63]. The effect of trial spacing on habituation remains poorly researched in invertebrates, and the sparse studies we found offer some conflicting results. For example, Thon [64] studied a visual startle response in blowflies and found that longer ITIs actually produced faster habituation, but the ITIs ranged from only 7 to 120 s. Several studies in visual startle responses in mud-flat crabs show either similar rates of habituation regardless of ITI, or more rapid habituation at lower ITIs, but these ITIs were also relatively short, ranging from 0 to 180 s [50,51]. Thon [64] suggests that his findings with blowflies likely differ from the expectation set by Thompson and Spencer [15,16] because very short ITIs may also cause sensitization, a learned increase in response, that occurs simultaneously with habituation, resulting in a change in overall response pattern. For more details on the relationship between habituation and sensitization, see [65]. It is likely that Thon’s explanation can be extended to other experiments using short ITIs, including some of the work with mud-flat crabs [50,51], that find different results than would be predicted by Thompson and Spencer [15,16]. In our case, the minimum ITI of five minutes was presumably large enough to eliminate the possibility of sensitization, so further increasing ITIs led to the expected decrease in rate of habituation. Unfortunately, we have no work with which to make direct comparison. Additional work using exceptionally short ITIs would also be beneficial.

Our experiment also found that the 1440 min (24 h) ITI did not cause any habituation. This is in line with our finding from experiment 2 that showed no retention of habituation in the second day of the procedure. If LSR can be maintained at a high level without habituation, then may also be possible to associate other stimuli, such as an odor, with the light-change stimulus, so that the new stimulus eventually also produces LSR. This associative conditioning would be similar to what has been done with feeding responses in other cockroach work [44,63]. However, if LSR habituates, as it does at lower ITIs, it may be challenging to associate a novel stimulus with a continually decreasing response. Future associative conditioning work may therefore consider using longer ITIs.

## 6. General Discussion and Conclusions

Our research is the first to describe the light-startle response (LSR), habituation of LSR, and any form of learning in the orange head cockroach (*Eublaberus posticus*). This work lays a strong foundation for future LSR habituation research on a variety of topics including other established principles of habituation [15,16], the relationship between habituation and sensitization [65], the potential differences between short-term and long-term habituation [55], and the effects of very small ITIs [64]. Research on individual differences may also be useful. With respect to individual weight, our work found one significant effect (experiment 1), one nearly significant effect (experiment 4), and two clearly nonsignificant effects (experiments 2 and 3). While it would be premature to speculate on what, if any, effect weight has on LSR, our results suggest future considerations of weight and other individual factors may be worthwhile. For any further research, development of new, more sophisticated methods may also be useful. While our experiments provide a solid basis for additional research, highly controlled, automated methods may reduce influence of extraneous factors, like human observation, and permit more detailed measurements, such as duration or distance of startle-induced movement. Additionally, the study of non-associative habitation conditioning may also provide a good framework for research in associative classical or operant conditioning [44,63]. A clear understanding of behavior and learning also creates possibilities to investigate the effects of various pharmacological or neurobiological manipulations on insect learning [37,66,67].

Another consideration is use of cockroaches in the classroom. The inexpensive, practical nature of insects makes them ideal organisms to demonstrate basic principles of learning in hands-on classroom exercises. While this idea is not new [68], there has been a recent emergence of laboratories using cockroaches as a medium to teach principles of comparative psychology and behavior analysis in the United States, including Mark Dixon’s laboratory at Southern Illinois University [69], Ben Witts’s laboratory at St. Cloud State University, Paul Andronis’s laboratory at Northern Michigan University, and Darby Proctor’s laboratory at the Florida Institute of Technology [70]. Our LSR conditioning methods offer a relatively practical, low-cost opportunity to demonstrate learning in classroom laboratories that may benefit many students.

We hope that future work considers LSR in the orange head cockroach for both research investigations and teaching demonstrations. While subsequent efforts using this under-represented cockroach (compared to *Periplaneta americana* and *Blattella germanica*) will be beneficial given the general lack of species diversity, and especially insect diversity, in comparative psychology [10], it is also important to continue to study a variety of species as notable differences can be found among related taxa. For example, Tomsic, Massoni, and Maldonado [71] found differences in startle habituation between related species of crabs with different ecologies. Finally, as we move forward in our understanding of comparative insect behavior and learning, we hope that researchers will consider the importance of publishing challenging papers regarding null findings and results of less exciting topics. Such publications are important to reduce the “file-drawer effect” sometimes observed in behavioral research [72,73]. For a comparative understanding, it is important to not only know what interesting things an insect can do, but also mundane findings and what insects cannot do.

## Figures and Tables

**Figure 1 insects-12-00339-f001:**
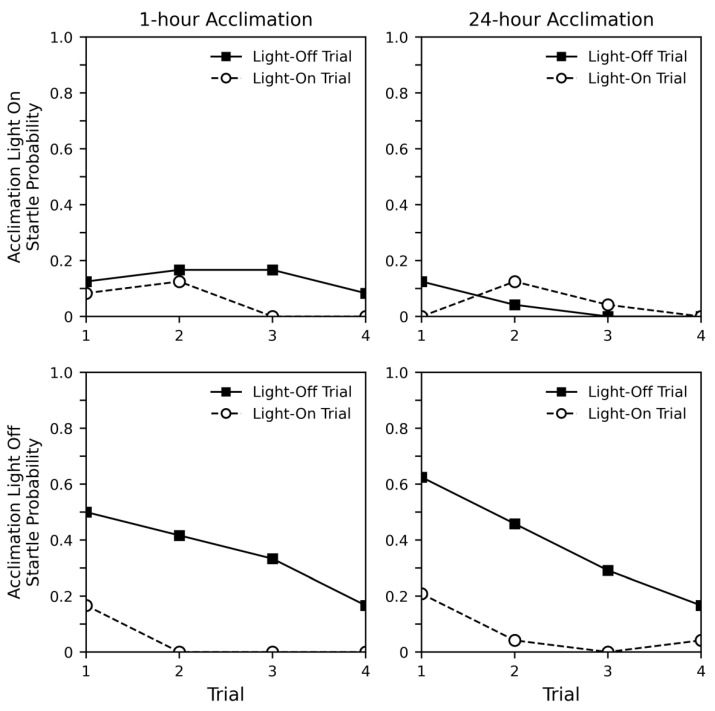
Probability of startle in response to types of light changes in experiment 1. All subjects showed a startle response when recollected after the final trial.

**Figure 2 insects-12-00339-f002:**
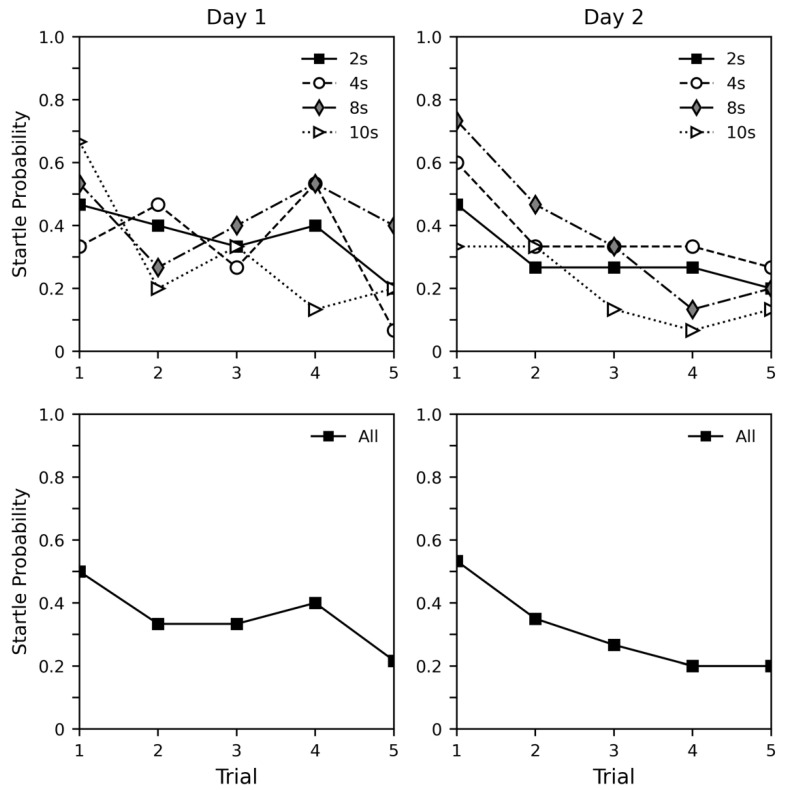
Probability of startle in response to light change in experiment 2. All subjects showed a startle response when recollected after the final trial. The top row shows the data split by stimulus duration. The bottom row shows all data pooled together.

**Figure 3 insects-12-00339-f003:**
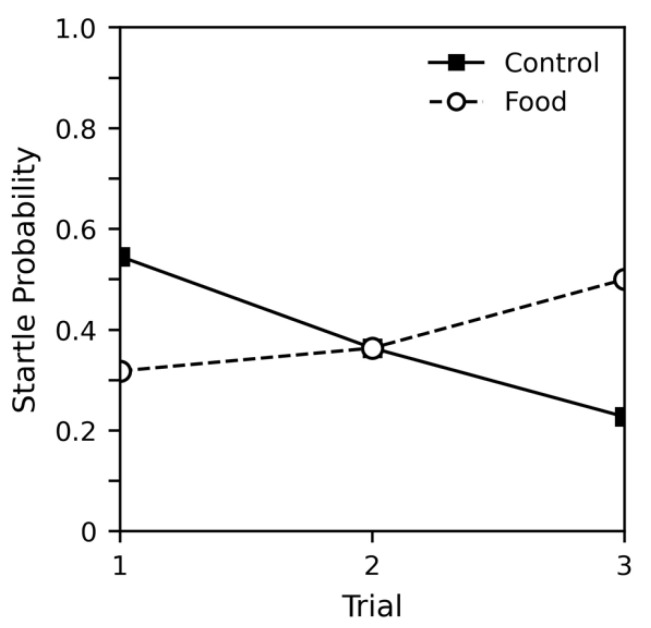
The effect of food presentation on probability of startle in response in experiment 3. All subjects showed a startle response when recollected after the final trial.

**Figure 4 insects-12-00339-f004:**
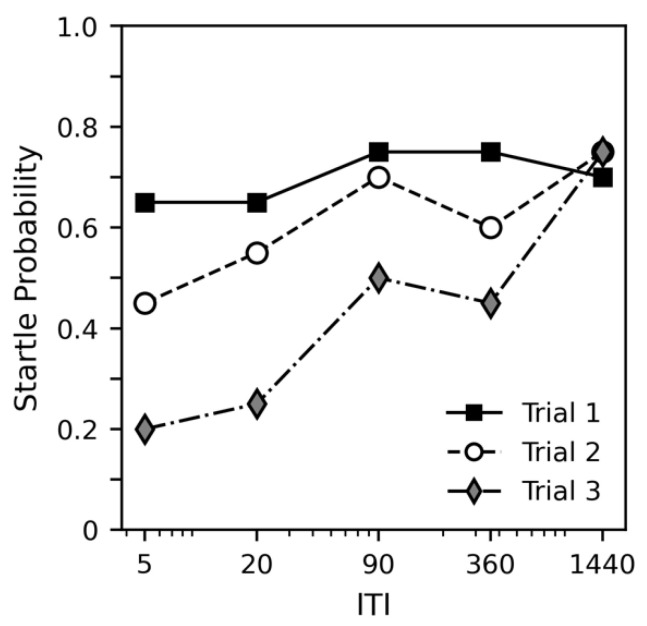
The effect of intertrial interval on rate of startle habituation in experiment 4. Intertrial interval is graphed on a logarithmic scale. All subjects showed a startle response when recollected after the final trial.

**Table 1 insects-12-00339-t001:** Experiment 1 Startle Probability Regression Analysis.

Parameter	Estimate	Standard Error	*p*-Value
Intercept	0.117	0.473	0.805
Light-On Trial	−1.918	0.689	0.005
Acclimation Light On	−2.839	0.642	0.000
Light-On Trial × Acclimation Light On	1.821	1.049	0.083
Trial	−0.309	0.060	0.000
Trial × Light-On Trial	−0.380	0.332	0.252
Trial × Acclimation Light On	0.136	0.136	0.319
Trial × Light-On Trial × Acclimation Light On	0.276	0.370	0.455
Weight (g)	0.352	0.137	0.010

Note. Confidence intervals are not shown to improve readability but can be derived from the parameter estimates and standard error.

**Table 2 insects-12-00339-t002:** Experiment 2 Startle Probability Regression Analysis.

Parameter	Estimate	Standard Error	95% Confidence Intervals		*p*-Value
Intercept	−0.057	0.512	−1.062	0.947	0.911
Trial	−0.222	0.078	−0.375	−0.069	0.004
Day	0.288	0.342	−0.382	0.957	0.400
Trial × Day	−0.180	0.104	−0.384	0.024	0.083
Light duration (s)	−0.014	0.048	−0.108	0.080	0.771
Weight (g)	0.063	0.064	−0.063	0.189	0.328

**Table 3 insects-12-00339-t003:** Experiment 3 Startle Probability Regression Analysis.

Parameter	Estimate	Standard Error	95% Confidence Intervals		*p*-Value
Control	1.121	1.126	−1.085	3.328	0.319
Food	−0.929	1.298	−3.473	1.615	0.474
Control × Trial	−0.704	0.273	−1.239	−0.170	0.010
Food × Trial	0.387	0.264	−0.130	0.904	0.142
Weight (g)	1.121	1.126	−1.085	3.328	0.319

**Table 4 insects-12-00339-t004:** Experiment 4 Startle Probability Regression Analysis.

Parameter	Estimate	Standard Error	95% Confidence Intervals		*p*-Value
Intercept	2.995	0.760	1.506	4.485	0.000
Trial	−0.838	0.154	−1.139	−0.536	0.000
ITI	−0.001	0.001	−0.002	0.001	0.260
Trial × ITI	0.001	0.000	0.000	0.001	0.004
Weight (g)	−0.353	0.183	−0.713	0.006	0.054

## Data Availability

The data presented in this study are available in supplementary material.

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
