# Peer review of "Habituation of the Light-Startle Response of Orange Head Cockroaches (Eublaberus posticus): Effects of Acclimation, Stimulus Duration, Presence of Food, and Intertrial Interval"

_insects, 2021, doi:10.3390/insects12040339_

Round 1

Reviewer 1 Report

Dear authors,

your present study is the first to describe the light-startle response (LSR), habituation of LSR, and any form of learning in the orange head cockroach (Eublaberus posticus). Habituation is a simple form of learning, thus, not less important compared to associative learning. As you mention, the results may lay a strong foundation for future LSR habituation research on a variety of topics including other established principles of habituation, the relationship between habituation and sensitization, the potential differences between short-term and long-term habituation, and the effects of very small ITIs. Additionally, the study of non-associative habitation conditioning may also provide a good framework for research in associative classical or operant conditioning. A clear understanding of behavior and learning also creates possibilities to investigate the effects of various pharmacological or neurobiological manipulations on insect learning.

For me, your experiments are well designed, the general idea behind each experiment, as well as the methods and results are clearly written and nicely discussed. I only have suggestions for some small minor revisions:  

Line 149: “While we did observe changes in antenna activity, primarily a cessation of antennae movement on stimulus presentation, we could not reliably observe this response and thus only considered the overall location of subjects.”

For me, this sentence is not clear. Does this count for the animals which already moved? Or all animals? And what do you mean by, you only considered the overall location of subjects?

Line 229: …this distinction

                   What do you mean by distinction in this case? Would the term “difference” not be better in this case, since you refer to more responses in the one, compared to the other case?

Line 235:  I am not sure you can argue like this. As you mention above, both, on-switch- and off-switch could indicate the presence of a potential predator…

Author Response

Thanks for your efforts. We responded to all comments. Please see the attachment for details.

Reviewer 2 Report

General comments:

This study investigated the habituation of the LSR behavior in orange head cockroaches, and tested the influences of different factors, such as acclimation, food, light duration, and intertrial interval. The writing is clear, but there are some concerns needs to be addressed prior to publication:  

Major comments:

  1. It is not clear how odor, sound, and human disturbances were avoided during all the experiments, and how these factors will influence the results and interpretation. And how the movements of the insects were recorded during the experiments, by camera, or human (which will introduce human disturbances)? A picture of the set up will be helpful. 
  2. Except experiment 1, which is repeatedly confirmed by experiment 2/3/4, it is not clear if there is experimental replication for experiment 2, 3, and 4.
  3. Weight was observed as a significant influencing factor in experiment 1 but not in experiment 2, this needs to be fully discussed.
  4. The structure of the paper was laid out by experiments, which is an uncommon structure of scientific papers of this kind, however it is not hard to follow, I will leave this for the editor to comment.

Other comments:

Line 54: “we believe that cockroaches are an ideal taxon…” please add some more explanations on why you think they are ideal, maybe from biological or evolutionary perspectives?

Line 348: “Table 2” to “Table 3”?

Line 367: “…during the course of the experiment, ”, it is unclear how long is the duration?

Author Response

Thanks for your efforts. We responded to all the comments. Please see the attachment for details.

Round 2

Reviewer 2 Report

I don't have any further comments.